# The Life of MicroRNAs: Biogenesis, Function and Decay in Cancer

**DOI:** 10.3390/biom15101393

**Published:** 2025-09-30

**Authors:** Shuang Ding, Pingping Wang

**Affiliations:** College of Life Sciences, Shandong Normal University, Jinan 250000, China; 2020010100@stu.sdnu.edu.cn

**Keywords:** miRNA biogenesis, Argonaute, TDMD, miRNA modification, tsRNA

## Abstract

MicroRNAs (miRNAs) are small non-coding RNAs that play pivotal roles in post-transcriptional gene regulation, influencing development, differentiation, and disease pathogenesis. Since their discovery in 1993, miRNAs have been recognized for their evolutionary conservation and pleiotropic effects, with the 2024 Nobel Prize underscoring their significance in post-transcriptional regulation via the RNA interference (RNAi) pathway. This review synthesizes the complete life cycle of miRNAs—from transcription and processing to function and decay—emphasizing regulatory mechanisms and their implications in human diseases, particularly cancer. We discuss how epitranscriptomic modifications influence miRNA biogenesis and activity, explore their nuclear and mitochondrial functions, and address emerging challenges in miRNA-based therapeutics, including the expanding small RNA landscape such as tRNA-derived small RNAs (tsRNAs), and Argonaute (AGO)-independent activities. Despite hurdles such as modest multi-target effects, off-target interactions, and delivery challenges, miRNAs remain promising as both biomarkers and therapeutic agents, underscoring the need for sustained research to bridge preclinical insights with clinical applications.

## 1. Introduction

MicroRNAs (miRNAs) were first identified in 1993 by Victor Ambros and Gary Ruvkun in *Caenorhabditis elegans* [1]. Initially overlooked as species-specific oddities, their importance became evident with the discovery of homologous sequences across species, revealing their evolutionary conservation and essential function in post-transcriptional gene regulation [2,3]. These ~22-nucleotide (nt) non-coding RNAs are now established as master regulators of gene expression, modulating processes such as development, cellular differentiation, and homeostasis [3].

The profound biological and clinical impact of miRNAs was highlighted by the 2024 Nobel Prize in Physiology or Medicine [4], following the 2006 award for the discovery of RNAi. However, miRNA biology remains complex, characterized by pleiotropic effects, context-dependent functions, and unresolved therapeutic challenges. Despite their short length, miRNAs exhibit distinct functional regions: they typically begin with a 5′-phosphorylated uridine, followed by a seed sequence spanning nucleotides 2–8 (with possible variations down to 5–6 nt) that is complementary to target RNAs [5]. MiRNAs sharing the same seed belong to families, often regulating overlapping targets. The seed is followed by a central bulge and a 3′-supplementary sequence that partially pairs with targets, influencing regulatory efficacy [5]. Variations in base pairing lead to diverse regulatory outcomes, positioning each miRNA as a three-dimensional entity whose interactions dictate specific structures and effects.

Extensive studies—from genetic screens to animal models [6,7]—have implicated miRNAs in human diseases, including cancer [8], neurodegeneration [9], and autoimmune disorders [10]. Dysregulation at any biogenesis stage—transcription, processing, export, loading, or decay—can contribute to pathogenesis. This review integrates recent advances across the miRNA life cycle, focusing on novel regulatory mechanisms and disease implications, especially in oncology. We highlight structure-specific determinants beyond sequence alone, explore epitranscriptomic layers, and address translational opportunities and barriers in miRNA therapeutics.

## 2. Transcription of MiRNA

Approximately half of all currently identified miRNAs are intragenic—located within introns (and occasionally exons) of protein-coding genes—and are typically co-transcribed with their host genes [11,12]. MiRNA hairpins are highly enriched in intronic regions, suggesting that introns serve as evolutionary hotspots for miRNA emergence [11].

The remaining miRNAs are intergenic and transcribed independently under the control of their own promoters. Some miRNAs are organized into genomic clusters and are transcribed as a single long primary transcript (pri-miRNA), often sharing seed sequences and forming functional families [13], such as the miR-17-92 cluster [14] or the miR-106a-363 cluster. MiRNAs from the same family can also be distributed across different gene loci, such as the let-7 family, whose members are located on seven different chromosomes [15].

Most miRNAs are transcribed by RNA polymerase II (Pol II) and, like other Pol II transcripts, carry a 5′ cap and a 3′ poly(A) tail [16,17]. However, a subset—including miR-515-1, miR-517a, and miR-519a-1 within the chromosome 19 miRNA cluster—is transcribed by RNA polymerase III [18]. Regardless of polymerase origin, all pri-miRNAs undergo multistep processing to become mature, functional miRNAs [16].

Numerous studies report that overall miRNA expression is downregulated in tumor tissues compared to normal counterparts [8,19,20,21], suggesting that disruptions in miRNA biogenesis—including transcriptional defects—may contribute to oncogenesis. While miRNAs can act as oncogenes or tumor suppressors depending on their cellular context and target genes, their expression is generally regulated at the transcriptional level, where they are initially generated.

### 2.1. Transcriptional Regulation

The transcription of miRNA genes is precisely controlled by transcription factors (TFs) and epigenetic modifications. Over 2600 human miRNAs have been cataloged (e.g., in miRbase as of 2025) with highly tissue- and cell-type-specific expression patterns [22,23]. Approximately 10% of miRNAs are expressed across most tissues, but none are uniformly expressed or considered housekeeping [23]. Another ~10% of miRNAs are highly tissue-specific, expressed primarily or exclusively in specific cell types [23]. Additionally, certain miRNAs predominate in specific tissues, constituting a significant portion of the total miRNA pool [24], such as miR-290–295 in embryonic stem cells (ESCs) [25,26], miR-122 in liver [24,27], and miR-1 in muscle [24,28].

Tissue specificity of miRNA expression is primarily determined by tissue-specific TFs [29,30], such as MYOD, REST and OCT4. These TFs are critical for development and cell differentiation through regulating miRNA transcription. For example, MYOD induces muscle cell differentiation by transcriptionally upregulating miR-1, miR-133, and miR-206, which target downstream genes such as PAX3 and UTRN, inhibitors of muscle differentiation [31,32]. REST, a repressor of neuronal genes, inhibits the transcription of brain-specific miRNAs like miR-124a, miR-9, and miR-132 in non-neuronal cells, influencing neuronal identity [33,34]. In pluripotent stem cells, OCT4, SOX2, and KLF4 suppress differentiation-promoting miRNAs such as miR-145 to maintain self-renewal [35,36].

The transcription of miRNAs is under epigenetic control. Both DNA methylation and histone modifications modulate miRNA promoter activity [37]. Super-enhancers (SEs) are large clusters of enhancers that play a crucial role in gene expression by recruiting high levels of TFs and co-activators like the Mediator complex [38]. They are characterized by epigenetic modifications such as the acetylation of histone H3 at lysine 27 (H3K27ac) and methylation of H3K4 (H3K4me1) and play a central role in maintaining cancer cell identity [39,40]. SE-associated miRNAs dominate ~60% of the miRNA pool in mouse ES cells, while typical enhancer-associated miRNAs account for ~20% [38]. Alterations in epigenetic modification, especially in the SE region, can significantly alter miRNA profiles and drive a cellular identification switch (Figure 1A).

### 2.2. Mutations of Transcriptional Regulators

Genetic mutations in miRNA loci or their transcriptional regulators can profoundly perturb miRNA expression and contribute to diseases, particularly cancer. Although most tumor genome studies have focused on protein-coding genes, accumulating evidence shows that mutations in miRNA genes and their regulatory factors also play critical roles in tumor initiation and progression [41].

Deletions or point mutations in miRNA loci directly impair expression. For instance, miR-15a and miR-16-1 at 13q14 are deleted in approximately 70% of chronic lymphocytic leukemia cases, leading to dysregulation of pro-survival genes [42].

More commonly, mutations or dysregulation in TFs and epigenetic regulators broadly affect the miRNA landscape. Mutations in MYC are associated with altered miRNA expression in various tumors. MYC activates the miR-17-92 cluster to promote survival and self-renewal [43] while repressing tumor-suppressor let-7 family members by binding upstream of the let-7a-1/let-7f-1/let-7d cluster [44]. The tumor-suppressor p53 upregulates the miR-34 family, which targets oncogenes affecting cell proliferation, apoptosis, and metastasis [45,46]. ZEB1 and ZEB2, key activators of epithelial–mesenchymal transition (EMT), repress the miR-200 family, which in turn targets EMT-associated genes including ZEB1/2, SNAIL, and VIM. Downregulation of the miR-200 family is associated with increased tumor aggressiveness and poor patient survival, underscoring its potential as both a biomarker and a therapeutic target [47] (Figure 1A).

Epigenetic dysregulation can drive cancer development even in the absence of driver mutations [48] and can also alter the expression of cancer-related miRNAs. Tumor-suppressor miRNAs like miR-124a in acute lymphoblastic leukemia can be silenced by aberrant CpG island methylation adjacent to their promoters [49]. And the frequency of human miRNA gene methylation is ~10 times higher than that of the protein-encoding genes [50]. Epigenetic modifiers such as HDACs and TET enzymes are often mutated or dysregulated in cancer [51]. HDACs are overexpressed in chronic lymphocytic leukemia, silencing miR-15a, miR-16, and miR-29b; HDAC inhibition restores their expression via the accumulation of the transcriptionally activating chromatin modification H3K4me2 in ~35% of patients [52].

Environmental chemicals induce epigenetic modifications and alter miRNA profiles. Heavy metal exposure (e.g., arsenic, cadmium) changes DNA methylation and miRNA expression, contributing to cancer development. For example, miR-146a upregulation is linked to cadmium exposure in steel workers, increasing their risk of cancer and cardiovascular disease [53]. The reversible nature of these epigenetic alterations, combined with the widespread impact of miRNA dysregulation, which positions them as promising therapeutic targets [51], as demonstrated by emerging strategies like HDAC inhibition to restore tumor-suppressor miRNAs.

In summary, miRNA transcription is controlled by a complex interplay of TFs and epigenetic modifications. Mutations or dysregulation at these levels can alter miRNA expression, contributing to oncogenesis. The high tissue specificity of miRNA transcription also makes them valuable biomarkers and potential therapeutic targets in cancer.

## 3. Processing in the Nucleus

Following transcription, primary miRNAs (pri-miRNAs) undergo essential maturation steps within the nucleus. These pri-miRNAs are typically long transcripts containing one or more stem-loop structures, which are recognized and cleaved by the Microprocessor complex—a minimal unit composed of the RNase III enzyme DROSHA and the double-stranded RNA-binding protein DGCR8 [16] (Figure 1B).

MiRNA processing by the Microprocessor complex is called the canonical pathway, which produces the majority of miRNAs. In addition to this dominant route, several non-canonical pathways contribute to miRNA biogenesis. These include mirtrons, which account for approximately 15% of human miRNAs and originate from spliced introns, bypassing Microprocessor complex processing [54,55], as well as simtrons, which are also generated from spliced introns but require only DROSHA for processing [56]. Despite the existence of alternative pathways, the canonical route remains predominant as knockout of DGCR8 results in more than a 95% global reduction in mature miRNAs in the skin cells, highlighting the essential role of the Microprocessor [57].

### 3.1. The Microprocessor Complex

The Microprocessor complex initiates pri-miRNA maturation in the nucleus. DGCR8 first binds to the stem-loop region of the pri-miRNA and recruits DROSHA, which cleaves the RNA to produce a ~70 nt precursor miRNA (pre-miRNA) with a characteristic 2-nt 3′ overhang. This cleavage event represents a key commitment step in miRNA biogenesis [16].

Structurally, the Microprocessor is a ~364 kDa heterotrimer composed of one DROSHA and two DGCR8 subunits. DROSHA (~160 kDa) serves as the catalytic core and contains multiple functional domains: an N-terminal proline- and RS-rich region for nuclear localization, a central conserved domain including a platform and PAZ-like domain, a connector region, two RNase III domains (RIIIDa and RIIIDb), and a C-terminal double-strand RNA-binding domain (dsRBD). These elements allow DROSHA to precisely recognize the basal junction of the pri-miRNA and cleave ~11 base pairs (bp) from the base and ~22 bp from the apical loop [16,58,59].

DGCR8 (~86 kDa per monomer) stabilizes and activates DROSHA. It contains a heme-binding domain (RHED), which is essential for dimerization and activity, as well as two dsRBDs that recognize the conserved UGU motif in the apical loop. DGCR8 also directly interacts with DROSHA’s C-terminal region to ensure precise positioning of the cleavage site [59].

Beyond its core components, some proteins are considered auxiliary components of the Microprocessor complex. For instance, hnRNP A1 facilitates the processing of miR-18a from the polycistronic miR-17–92 cluster by binding to the terminal loop of pri-miR-18a and inducing structural changes that make the cleavage site more accessible to DROSHA [60]. The efficiency of processing is also influenced by specific RNA sequence motifs (e.g., UG, CNNC, UGUG) and secondary structure, though most human pri-miRNAs lack all optimal features [16,61]. These limitations are often compensated for by regulatory RNA-binding proteins (RBPs), such as QKI5 and NONO, which modulate the processing of certain miRNAs depending on their binding preferences [62].

### 3.2. Regulation of Nuclear Processing

The activity of the Microprocessor complex is finely regulated at multiple levels, including transcription, complex assembly, autoregulatory loops, and post-translational modifications (PTMs). A feedback mechanism exists between DROSHA and DGCR8: DROSHA cleaves a hairpin structure in the 5′ UTR of DGCR8 mRNA to reduce its expression, while DGCR8 stabilizes DROSHA protein levels. This reciprocal regulation ensures homeostatic control of Microprocessor complex activity [63].

RBPs can both promote and inhibit miRNA processing in the nucleus [62]. NF90 can inhibit the processing of certain miRNAs by binding to pri-miRNAs and sterically hindering the Microprocessor’s access to their cleavage sites [64]. And in some cancers, dysregulation of certain RBPs can alter the processing efficiency of miRNAs. LIN28 is overexpressed in various cancers, promoting cell growth and proliferation by blocking the processing of let-7 precursors into mature, functional miRNAs [65].

PTMs play a crucial role in tuning the activity and stability of Microprocessor components. For DROSHA, phosphorylation at Ser300/Ser302 by GSK3β and PLK1 enhances nuclear localization, promotes DGCR8 binding, and increases the catalytic activity of miRNAs such as miR-1248, miR-1306-5p, and miR-2277-5p [66,67]. Acetylation protects DROSHA from ubiquitin-mediated degradation and is enhanced by HDAC inhibitors, such as Trichostatin A and nicotinamide, which also increase the expression of miRNA-143 [68]. Conversely, ubiquitination by the E3 ligase MDM2, downstream of mTORC1 signaling, targets DROSHA for degradation, leading to a global reduction in miRNA levels [69] (Figure 1B).

DGCR8 is likewise subject to multilayered regulation. Phosphorylation at Tyr267 by ABL kinase enhances its processing activity of miR-34c in response to DNA damage [70]. SUMOylation at Lys707 stabilizes DGCR8 and improves its pri-miRNA binding capacity, promoting cancer cell migration and transformation [71]. Moreover, DGCR8 can bind heme via its RHED. Heme binding is essential for DGCR8′s proper structural conformation and productive interaction with DROSHA [72,73] (Figure 1B).

Mutations in DROSHA or DGCR8 impair Microprocessor complex function, leading to a global reduction in miRNA levels in various cancers [74]. While some mutations can also alter their processing preference, leading to increased expression of oncogenic miRNAs [75]. DROSHA and DGCR8 can be either upregulated or downregulated in cancers, often depending on the cancer type, which suggests the tissue- and cell-specificity of miRNAs and the diverse roles they play in cancer development [8].

In summary, the Microprocessor complex functions as a central regulatory hub in miRNA biogenesis. Its activity is modulated by intrinsic structural domains, PTMs, binding partners, and homeostatic feedback. Disruption of any of these mechanisms can profoundly reshape the miRNA landscape, often promoting cancer progression.

### 3.3. Nuclear Export of Pre-miRNA

Following nuclear processing, pre-miRNAs with a 2-nt 3′ overhang and 5′ monophosphate are exported to the cytoplasm by Exportin-5 (XPO5) both through Ran-GTP–dependent and independent mechanisms [76]. XPO5 recognizes stem-loop structures longer than 16 bps. While variations at the 3′ end are tolerated, the presence of a 5′ overhang significantly impairs binding and export, underscoring the importance of precise Microprocessor cleavage [77,78].

Although XPO5 is not strictly essential for all pre-miRNA export, its knockout leads to a notable, albeit partial, reduction in mature miRNA levels [79]. This suggests some degree of functional redundancy, potentially involving other RNA-binding proteins or alternative exportins.

XPO5 activity is modulated by upstream signaling pathways. The ATM/AKT pathway enhances pre-miRNA export by promoting phosphorylation of NUP153, a nuclear pore protein that interacts with XPO5 [80]. In contrast, ERK-mediated phosphorylation of XPO5 inhibits its function, resulting in widespread suppression of miRNA maturation [81]. Importantly, XPO5 expression is frequently dysregulated in cancer and may function either as an oncogene or tumor-suppressor, depending on the cellular context and cancer type [82] (Figure 1C).

In summary, nuclear processing and export of miRNAs are complex and tightly regulated events. The Microprocessor complex, accessory RNA-binding proteins, and nuclear export machinery together ensure efficient and accurate miRNA maturation. Disruptions at any stage of this process can lead to aberrant miRNA profiles and contribute to pathogenesis, particularly in cancer.

## 4. Processing in the Cytoplasm

After export from the nucleus, pre-miRNAs enter the cytoplasm, where they undergo final processing steps essential for producing mature, functional miRNAs. This stage is primarily orchestrated by the RNase III enzyme DICER and its cofactors, which generate a ~22 nt miRNA duplex that is subsequently loaded into AGO proteins to form the RNA-induced silencing complex (RISC). The precise processing of pre-miRNAs at this step critically determines the length, sequence, and strand selection of mature miRNAs, influencing their stability and gene regulatory function.

### 4.1. DICER-Mediated Cleavage

DICER is a large, multi-domain endoribonuclease responsible for converting pre-miRNAs into miRNA duplexes. It recognizes the characteristic hairpin structure of pre-miRNAs, which typically contain a 2-nt 3′ overhang and a ~70 nt stem-loop [16] (Figure 2A).

The enzyme’s structure enables highly specific substrate recognition and precise cleavage. The N-terminal DExD/H-box helicase domain engages the terminal loop of the pre-miRNA, clamping and positioning the RNA for accurate processing. The adjacent DUF283 domain, structurally resembling dsRBD, may assist in binding single-stranded RNA regions and contribute to substrate recognition. The PAZ domain binds the 3′ overhang of the pre-miRNA, anchoring it firmly and functioning as a molecular ruler that measures a fixed distance (~22 nt) from the end to the cleavage site. Two RNase III domains (IIIa and IIIb) form a dimer that catalyzes staggered cuts on each RNA strand, generating the characteristic 2-nt 3′ overhang on the miRNA duplex. Finally, a C-terminal dsRBD further stabilizes the interaction with dsRNA substrates, enhancing cleavage fidelity. The spatial arrangement of these domains ensures DICER cleaves pre-miRNAs at consistent positions, producing duplexes of defined length crucial for downstream strand selection and function [83,84].

Structural variations within pre-miRNAs, such as differences in terminal loop size, 3′ overhang length, and internal bulges or mismatches, influence DICER’s substrate recognition and cleavage precision [83,85]. These features can create subtle shifts in cleavage sites, producing isomiRs—miRNA variants with sequence or length differences—which expand the regulatory repertoire of miRNAs [86].

The biological importance of DICER is supported by strong genetic evidence. *Dicer1* knockout mice exhibit embryonic lethality, while conditional knockouts display various developmental defects [87,88,89,90]. Somatic mutations in the *Dicer1* gene, especially in the RNase IIIb domain, have been identified in several human cancers, including Wilms tumor, pleuropulmonary blastoma, and others [8,91]. These mutations disrupt processing of miRNAs, skewing miRNA populations and contributing to oncogenesis. Beyond cancer, altered *Dicer* expression or mutations are implicated in neurodegenerative diseases, metabolic disorders, and developmental defects, emphasizing the enzyme’s broad physiological importance [92].

Moreover, DICER’s activity and stability are dynamically regulated by PTMs. Phosphorylation via signaling pathways like ERK and mTORC2 can modulate DICER’s localization and catalytic efficiency, linking miRNA processing to extracellular cues and cellular states [93,94]. Ubiquitination, SUMOylation and glycosylation also regulate DICER turnover, allowing cells to finely tune miRNA production in response to stress, differentiation, or disease [95,96] (Figure 2A).

### 4.2. Regulation of Cytoplasmic Processing by TRBP and PACT

DICER does not act alone. It forms complexes with cofactors such as TRBP and PACT, which bind primarily to the helicase and DUF283 domains. These cofactors stabilize the DICER-pre-miRNA complex, enhance cleavage efficiency, and can influence cleavage site selection, thereby affecting miRNA isoform diversity [97,98,99].

TRBP contains three dsRNA-binding domains, although only the first two are primarily responsible for RNA binding. It enhances DICER’s catalytic efficiency by stabilizing the interaction between DICER and its pre-miRNA substrates [97]. It also serves as a molecular bridge facilitating the transfer of the miRNA duplex from DICER to AGO proteins, promoting efficient RISC assembly [100]. Beyond catalysis, TRBP modulates substrate selectivity, fine-tuning miRNA expression profiles [101,102]. TRBP’s role in miRNA processing also links it to antiviral defense mechanisms, as it interacts with the kinase PKR—a key sensor of dsRNA—thereby inhibiting PKR activation and preventing excessive cellular stress responses [103].

PACT, also a dsRBP with three dsRNA-binding domains, forms a functional complex with DICER and supports its catalytic activity [98]. Structural studies show that PACT binds to DICER in a manner similar to TRBP, but through mutual exclusion, leading each to selectively promote miRNA processing [97]. At the same time, PACT is a known activator of PKR, promoting antiviral and stress-response signaling pathways. Within the cell, TRBP and PACT compete for PKR and DICER binding, creating a finely balanced system that coordinates stress and immune signaling with miRNA processing [104].

Interestingly, the DICER–TRBP and DICER–PACT complexes differentially affect miRNA biogenesis. For some pre-miRNA substrates, such as pre-miR-200a and pre-miR-34c, these complexes produce different isomiR patterns by varying cleavage sites. Experiments swapping their RNA-binding domains confirmed that these functional differences largely depend on their N-terminal dsRNA-binding domains. Moreover, TRBP generally promotes DICER stability and miRNA dicing, while PACT may inhibit processing under certain conditions, illustrating a nuanced regulatory interplay [99].

In addition to TRBP and PACT, many other RBPs have been reported to regulate DICER-mediated processing. For example, LIN28 binds to the terminal loop of pre-let-7, inhibiting DICER cleavage and promoting its degradation [105,106]. YB-1, which is often overexpressed in various cancers, binds the terminal loop of pre-miR-29b-2 and suppresses the tumor-suppressive effects of miR-29b [107].

Thus, the DICER cofactors TRBP and PACT serve as critical regulators of cytoplasmic miRNA maturation, linking RNA processing to cellular stress responses and expanding the functional diversity of the miRNA pool. In addition, many cytoplasmic RBPs can regulate the maturation of specific miRNAs and contribute to cancer development.

### 4.3. MiRNA Loading and Strand Selection

Once DICER generates the miRNA duplex, the next step is to load one strand—the guide strand—into an AGO protein to form RISC, while the other, known as the passenger strand (miRNA*), is typically degraded (Figure 2B).

Humans express four AGO proteins (AGO1–4), which share approximately 80% amino acid identity and have largely overlapping miRNA binding profiles [108,109]. These proteins have conserved domains (N-terminal, PAZ, MID, PIWI), forming a bilobed structure that interacts with miRNAs. AGO2 is unique in its catalytic “slicer” activity and critical role during early development [110], making it the most studied.

MiRNA loading is not a random process, and strand selection depends on two main principles: AGO preferences and the thermodynamic asymmetry of the duplex ends [16,109,111]. AGO proteins preferentially bind miRNA strands whose 5′ end is less stably paired [112]. The MID domain specifically anchors the 5′ phosphate of the guide strand and has a preference for uridine at this position [113], which further biases strand selection.

Thermodynamic asymmetry arises because the two ends of the miRNA duplex differ in base-pairing stability; the strand with the weaker 5′ end is more readily loaded into AGO. This asymmetric loading ensures that the correct strand guides target recognition [111].

However, strand selection is more complex than these rules alone. In different tissues or under various physiological or pathological states, arm switching can occur, where the dominant strand loaded into AGO changes [114]. This phenomenon is influenced by RBPs, post-transcriptional miRNA modifications [115], and cellular chaperones such as Hsc70 and Hsp90, which facilitate the ATP-dependent loading of miRNA duplexes into AGO [116,117].

Notably, cancer-associated miRNA mutations or altered expression of miRNA regulators frequently disrupt strand selection and promote arm switching, contributing to oncogenic deregulation [111,118]. For example, miR-574-5p and miR-574-3p are inversely expressed and play opposite roles in gastric cancer progression, with a high 5p/low 3p expression pattern significantly associated with advanced TNM stages and poor patient prognosis [119]. This dynamic regulation of miRNA strand loading adds another layer of control to miRNA-mediated gene silencing and highlights potential diagnostic and therapeutic targets.

The cytoplasmic processing of miRNAs involves tightly regulated steps from DICER-mediated cleavage to RISC assembly. The multi-domain enzyme DICER works with cofactors TRBP and PACT to generate precise miRNA duplexes. Subsequent strand selection and loading into AGO proteins involve complex recognition of structural features and cellular context, ensuring functional miRNAs guide gene silencing. Disruption of any of these steps leads to widespread dysregulation implicated in cancer, neurodegeneration, and other diseases. Understanding these processes offers insights into miRNA biology and potential avenues for therapeutic intervention.

## 5. Function of MiRNA in RISC

Once mature miRNAs are loaded into AGOs to form RISC—a multiprotein complex centered on AGOs and guide RNAs—they become potent regulators of gene expression. About 60% of the whole protein-coding gene population has at least one miRNA target site [120]. This percentage increases to over 80% among cancer-related genes [121]. The canonical role of RISC is to repress target mRNAs post-transcriptionally, primarily in the cytoplasm. However, recent advances have revealed diverse noncanonical functions extending beyond traditional mRNA silencing, including nuclear activities and epigenetic regulation. Moreover, mutations and PTMs of AGO proteins critically influence miRNA function and are increasingly linked to the development and progression of human diseases.

### 5.1. Canonical Function: Translational Inhibition and mRNA Destabilization

Although in many organisms the cleavage activity of AGOs functions in immunity or regulation of endogenous gene expression, these functions seem to be largely absent in mammals [122]. Even though human AGO2 and AGO3 retain their catalytic activity in miRNA processing, they rarely cleave target RNAs in vivo. Studies from other organisms and siRNA research have shown that target cleavage occurs only when the guide RNA and target exhibit near-perfect base pairing [123,124], even though recent high-throughput analyses suggest that the cleavage mechanism of human AGO2 is more complex than previously thought [125,126] and may require further investigation from a structural perspective of the complex, rather than viewing it solely as a simple linear base-pairing process. Notably, since miRNAs primarily regulate their targets through the seed sequence, their function is not heavily dependent on the cleavage activity of AGO.

In mammalian cells, miRNAs primarily function to repress gene expression by guiding RISC to partially complementary sequences typically located in the 3′ untranslated region (UTR) of target mRNAs [8,109]. This targeting leads to translational inhibition and mRNA destabilization, two intertwined mechanisms that together reduce protein output effectively (Figure 3A).

Translational inhibition can be achieved through several mechanisms [127], most notably by blocking the initiation phase of protein synthesis. RISC interferes with the assembly of the eukaryotic initiation factor 4F (eIF4F) complex and prevents ribosome recruitment, thus reducing the rate at which target mRNAs are translated into protein [128]. This mode of repression can be rapid and reversible, allowing cells to fine-tune protein production in response to stimuli.

Alongside translational repression, miRNA-loaded RISC recruits GW182 proteins, which serve as scaffolds to recruit deadenylase complexes such as CCR4-NOT. The deadenylation of the mRNA poly(A) tail triggers decapping and exonucleolytic degradation, thereby decreasing mRNA stability and reducing transcript abundance [127,129]. This dual action ensures both immediate and sustained downregulation of target gene expression.

Dysregulation of these canonical pathways is a hallmark of many cancers. Tumor-suppressor miRNAs like the let-7 family repress oncogenes such as RAS and MYC, and their loss contributes to unchecked cell proliferation [130,131]. Conversely, oncomiRs such as miR-21 target tumor-suppressor genes, including PTEN and TPM1, promoting tumor cell survival and growth. The balance of miRNA-mediated gene repression is thus critical for maintaining cellular homeostasis and preventing malignancy [132,133].

### 5.2. Functions in the Nucleus, Mitochondria and Condensate

Beyond their well-known cytoplasmic functions, miRNAs and AGO proteins also operate within the nucleus, exerting regulatory effects on transcription, chromatin modification, and RNA processing.

The nuclear localization of AGO is regulated both by AGO-binding proteins and by miRNA loading. Importins, particularly Importin-8, have been identified as mediators of AGO2 nuclear import [134], although redundant pathways may exist, as a significant portion of AGO remains nuclear even when Importin-8 is depleted. GW182, a key component in RISC translational regulation, shuttles between the cytoplasm and nucleus; its depletion leads to enrichment of AGO2 in the nucleus, while its expression retains AGO2 in the cytoplasm, thus influencing AGO subcellular localization [135]. MiRNA loading is essential for AGO-miRNA nuclear transport: in DROSHA knockout cells, AGO2 remains strictly cytoplasmic, but nuclear localization can be rescued by delivery of exogenous miRNAs—even negative-control siRNAs that do not target any human sequence. Nuclear localization of AGO-miRNA complexes is also affected by cell density, with nuclear AGO levels increasing under high confluency [136]. This may serve as a regulatory mechanism to relieve cytoplasmic RNAi activity in response to stresses such as external mechanical forces.

Nuclear AGOs, directed by miRNAs, can localize to gene promoters and enhancers to influence chromatin accessibility and transcriptional activity. Histone methyltransferase EHMT2, which suppresses the expression of fumarate hydratase in nasopharyngeal carcinoma, is recruited along with EZH2 by miR-584-3p in an AGO2-dependent manner to reduce matrix metalloproteinase 14 (MMP-14) expression in gastric cancer [137]. Endogenous nuclear miR-589 activates expression of COX-2 through the sense promoter RNA-mediated histone modification by WDR5 [138].

MiRNAs also serve as an enhancer trigger in the nucleus. MiR-24-1 can recruit transcriptional activators—RNA polymerase II, p300/CBP, and enhancer RNAs—to its neighboring genes, FBP1 and FANCI, through AGO2-dependent localization [139]. Many miRNAs have been found to influence the expression of their neighboring genes, highlighting their role in the cell-specific regulation of 3D chromatin organization [140].

MiRNAs also modulate alternative splicing by interacting with intronic sequences or splicing factors, thereby altering splice site selection and expanding proteomic diversity. AGOs have been detected binding intronic sequences, suggesting that nuclear RISC can associate with pre-mRNAs and regulate their splicing [141,142,143]. AGO3 can form a complex with SF3B3 to regulate pre-mRNA splicing, thereby restraining type 2 immunity [144]. In addition, miR-10b promotes glioma progression, in part, by binding U6 snRNA, a core component of the spliceosome, thereby affecting the splicing of genes such as CDC42 [145].

Besides nuclear RNAi, several miRNAs have also been identified in mitochondria, termed mitochondrial microRNAs (mitomiRs) [146]. MitomiRs such as miR-696, miR-532, miR-690, and miR-345-3p may play important roles in the early stages of heart failure [147]. MitomiRs such as miR-92a-2-5p and let-7b-5p contribute to impaired mitochondrial gene expression and increased ROS production. [148]. Since cardiomyocytes are rich in mitochondria, identifying the mitomiRs will aid in early and accurate diagnosis. This could facilitate effective therapy and reduce mortality.

RISCs often assemble into cytoplasmic non-membranous bodies, such as P-bodies and stress granules [149], which are driven by liquid–liquid phase separation with the assistance of other proteins [127]. In P-bodies, GW182-AGOs interactions promote phase separation and contribute to mRNA decay [150]. Under stress conditions (e.g., heat shock, nutrient deprivation, or viral infection), stress granules help cells survive adverse environments by temporarily storing some components and providing a site for RNA repair [151]. Due to the highly dynamic nature of biomolecular condensates [152], miRNAs can modulate mRNA structure, thereby facilitating the translation of specific transcripts (Figure 3B).

Advanced RNA detection technologies have enabled the precise identification of miRNA subcellular localization, expanding our understanding of RNAi. RNAi can occur in both the nucleus and cytoplasm, where distinct molecular environments may confer different functional mechanisms. Even within the cytoplasm, the discovery of phase separation has revealed that miRNAs can exert diverse effects on gene regulation in response to a variety of environmental stimuli.

### 5.3. Post-Translational Regulation of RISC in Diseases

The function of RISC is tightly controlled by genetic and post-translational mechanisms, and its disruption contributes to a wide range of diseases, including cancer, neurological disorders. Beyond AGOs, several other RISC components—such as GEMIN3/4 and GW182—are subject to mutations or dysregulation in pathological contexts [74].

Mutations in AGOs have been identified in some diseases, especially neurodevelopmental disorders [153,154,155] and their dysregulation is associated with various cancers [156,157]. For example, heterozygous mutations in AGO1 cause Neurodevelopmental Disorder with Language Delay and Behavioral Abnormalities, with or without Seizures (NEDLBAS). Similarly, mutations in AGO2 result in Lessel–Kreienkamp syndrome [158]. In addition to genetic mutations, abnormal expression of AGOs is widely observed in cancers. For example, AGO3 is upregulated in cervical cancer, where it promotes malignancy and supports tumor growth [159]. AGO2 directly interacts with the oncogene KRAS; elevated AGO2 levels enhance neoplastic transformation in KRAS-driven cancers, whereas AGO2 knockout results in growth arrest [160].

PTMs, particularly phosphorylation and ubiquitination, dynamically regulate AGO2 function under both physiological and pathological conditions. AGO2 undergoes a phosphorylation cycle on residues Ser824–834, regulated by CK1α and reversed by the Protein Phosphatase 6 complex (PP6). CK1α-mediated phosphorylation alters AGO2’s interaction with target mRNAs, reducing its ability to recruit the silencing machinery. This phosphorylation promotes target mRNA release and remodeling of RISC activity. Conversely, PP6 dephosphorylates AGO2, restoring target binding and silencing capacity. Together, the CK1α–PP6 cycle acts as a reversible switch that fine-tunes miRNA-mediated gene silencing in response to cellular signals. [161]. In addition, phosphorylation of neuronal AGO2 at Ser387 via AKT3 downstream of NMDA receptor activation enhances miRNA-mediated translational repression critical for synaptic plasticity and memory [162]. Hypoxia-induced ubiquitination of AGO2 is mediated by LUBAC, which catalyzes M1-linked ubiquitination that can be removed by OTULIN. This ubiquitination stabilizes mRNAs by interfering with miRNA-mediated silencing, thereby promoting tumor progression under low-oxygen conditions. Furthermore, AGO ubiquitination by ZSWIM8 can be triggered by the binding of miRNAs to specific target RNAs, a mechanism that will be discussed later (Figure 2B).

In addition to AGO-centered regulation, miRNA sponges represent another mechanism of controlling miRNA activity [163]. Competing endogenous RNAs (ceRNAs)—such as long non-coding RNAs (lncRNAs) and circular RNAs (circRNAs)—harbor multiple miRNA-binding sites and can sequester miRNAs away from their mRNA targets [164]. For example, circRNA circNRIP1 functions as a sponge for miR-149-5p, thereby derepressing AKT1 expression and promoting gastric cancer progression [165].

Collectively, genetic and post-translational regulatory mechanisms act as molecular switches that control AGO2’s interaction, activity, and localization, with significant implications for development, cancer, and neuronal function.

In summary, miRNAs exert their gene regulatory functions predominantly through the RISC by mediating translational inhibition and mRNA destabilization, which constitute their canonical roles essential for maintaining cellular homeostasis. Beyond these classical functions, miRNAs and AGO proteins participate in diverse noncanonical activities within the nucleus and mitochondria, including transcriptional and epigenetic regulation as well as RNA processing, expanding their influence in gene expression control and cancer biology. Moreover, mutations and dynamic PTMs of AGO proteins critically shape miRNA functionality, with profound impacts on disease processes such as tumor progression and neurological disorders. Understanding these multilayered mechanisms provides valuable insights into miRNA biology and offers promising avenues for therapeutic intervention.

## 6. MiRNA Decay

MicroRNAs undergo a complex lifecycle, starting from biogenesis to eventual degradation. While various biogenesis byproducts are generally rapidly degraded, mature miRNAs are relatively stable. Their longevity primarily results from a tight association with AGO proteins, which shield them from exonucleolytic attack and maintain their function within RISC. Nonetheless, miRNAs are not permanent cellular components; their turnover is controlled by regulated decay pathways in both the cytoplasm and nucleus, allowing dynamic modulation of gene expression.

### 6.1. Degradation of MiRNA Intermediates and Mature MiRNAs

During miRNA maturation, several non-functional intermediates are produced, including the cleaved 5′ or 3′ fragments of pri-miRNAs, the terminal loop of pre-miRNAs, and the passenger strand from the miRNA duplex. These RNA fragments are rapidly eliminated by cellular RNA degradation machinery [79,166], such as 5′-to-3′ exoribonuclease 1/2 (XRN1/2) [167,168], 3′-to-5′ exoribonucleases PNPT1 [169] and ERI1 [170], and the endoribonuclease Tudor-SN [171]. Prompt clearance prevents unwanted interactions with RBPs and maintains cellular RNA homeostasis (Figure 4A).

In contrast, mature miRNAs generally exhibit much longer half-lives, ranging from hours to days, owing to their stable incorporation into AGO proteins [166,172]. MiRNA half-lives are tightly regulated and display strong tissue specificity. MiRNAs are protected by AGO proteins, with their 5′ ends anchored in the MID domain, their internal regions shielded by surrounding AGOs to limit RNase accessibility, and their 3′ ends generally secured within the PAZ domain.

However, interactions with target RNAs or chemical modifications at the 3′ end can trigger conformational changes in RISCs, leading to the release of the miRNA 3′ end [126,173,174] and initiating its degradation through multiple mechanisms. The released 3′ end can undergo trimming by exonucleases or tailing by terminal nucleotidyl transferases. Terminal nucleotidyl transferases (TENTs) can modify the 3′ end of miRNAs, such as 3′ uridylation by TUT4/7, which promotes degradation by the U-specific exonuclease DIS3L2 [175].

Extensive base pairing between miRNAs and specific target RNAs can also trigger miRNA degradation through inducing AGO ubiquitination and degradation by proteasome, a process known as Target-Directed miRNA Degradation (TDMD), ultimately leading to miRNA release from AGO and degradation [174,176].

### 6.2. Target-Directed miRNA Degradation

TDMD occurs when target RNAs have extensive complementarity to the miRNA’s 3′ supplementary sequence in addition to the seed region. This extended pairing extracts the 3′ end of the miRNA from the PAZ domain and stabilizes a conformation of RISC that is distinct from that which occurs when the AGO–miRNA complex binds with less complementarity to a target [177]. ZSWIM8, a substrate receptor of a Cullin-RING E3 ubiquitin ligase (CRL), is able to recognize the RISC in TDMD and assemble with elongins B and C (ELOB and ELOC), cullin 3 (CUL3), ARIH1, and RBX1 to catalyze the transfer of an activated ubiquitin from an E2 ubiquitin-conjugating enzyme to AGO. Polyubiquitinated AGO is then degraded by the proteasome, releasing both the miRNA and its trigger RNA [178,179,180]. The released miRNAs are generally highly sensitive to RNases and rapidly degraded (Figure 4B).

TDMD is a conserved mechanism of miRNA decay from *C.elegans* to human [180]. A wide range of transcripts can serve as TDMD triggers and induce certain phenotypes by inhibiting miRNAs. The long non-coding RNA Cyrano directs efficient degradation of miR-7, thereby regulating neural activity [181]. Certain mRNAs, such as Serpine1, have been found to trigger the degradation of miR-30b/c [182], while BCL2L11 promotes miR-222 degradation, enhancing BIM-induced apoptosis in cancer cells [183]. Virus-derived snRNA-like RNA HSUR1 reduces T cell activation by triggering the degradation of miR-27a [176].

The importance of TDMD is underscored by genetic studies. ZSWIM8, an essential mediator of TDMD, is an evolutionarily conserved protein that plays a crucial role in regulating gene expression and development across various metazoans, from nematodes to humans. Knockout models of *Zswim8* exhibit severe developmental defects and perinatal lethality in mice, accompanied by aberrant accumulation of more than 50 miRNAs across 12 tissues [184,185]. Notably, deletion of two miRNAs, miR-322 and miR-503, can rescue growth of Zswim8-null embryos [185,186], illustrating the physiological relevance of tightly controlled miRNA degradation.

Although ZSWIM8 has been identified as a key factor in recognizing RISC during TDMD, the molecular features that target AGO for degradation remain to be elucidated, including how RNA–RNA pairing induces structural changes in AGO. Furthermore, additional studies are needed to catalog ZSWIM8 loss-of-function phenotypes and to determine the extent to which loss of TDMD contributes to these phenotypes.

## 7. RNA Modifications Add a New Regulatory Layer to MiRNA Biogenesis and Function

Recent research has uncovered a growing repertoire of chemical RNA modifications —collectively referred to as the epitranscriptome—that significantly influence miRNA biogenesis, stability, and function. Beyond well-characterized 3′ uridylation and adenylation, an expanding set of modifications—including adenosine-to-inosine (A-to-I) editing [187], N6-methyladenosine (m^6^A) [188], 5-methylcytosine (m^5^C) [189], 7-methylguanosine (m^7^G) [190], 2′-O-methylation [191], N4-acetylcytidine (ac^4^C) [192], 8-oxoguanine (8-oxoG) [193], 5′-O-methylation [194]—have been identified. As detection technologies continue to evolve [195], it is likely that many more miRNA modifications will be uncovered.

MiRNA modifications function as regulatory switches superimposed on the miRNA pathway, dynamically modulating gene silencing in response to developmental signals, cellular stress, and disease states [16,196]. As many as 16% of human pri-miRNAs are subject to A-to-I editing, which can have a significant impact on miRNA processing [187]. And at least 6% of mature miRNAs show A-to-I editing that could change miRNA binding specificity [197]. These modifications exhibit strong tissue specificity and play critical roles in development and tissue differentiation. Their dysregulation has been observed in various diseases, including cancer, where they contribute to disease onset and progression.

### 7.1. RNA Modifications During MiRNA Biogenesis

RNA modifications, both internal (within the nucleotide sequence, such as m^6^A or m^5^C) and terminal (such as 3′ uridylation or 2′-O-methylation at the 3′ end), regulate the efficiency and accuracy of miRNA maturation from primary transcripts to functional strands loaded into AGO proteins by fine-tuning their interaction with miRNA biogenesis regulators.

Adenosine deaminases acting on RNA (ADARs) catalyze A-to-I editing within double-stranded regions of pri- or pre-miRNAs. For example, editing of pri-miR-142 by ADAR1/2 inhibits DROSHA cleavage [198], while editing of pre-miR-151 prevents DICER processing [199]. Intriguingly, ADAR knockout selectively increases the levels of certain miRNAs, such as miR-142, because ADAR1 not only edits RNA but also interacts with DICER to promote miRNA processing through an editing-independent mechanism [200].

METTL3 enhances pri-miRNA processing by installing m^6^A marks on pri-miRNAs [188]. Mechanistically, distinct m^6^A reader proteins recognize these modified pri-miRNAs and recruit the Microprocessor complex to promote their cleavage—for example, hnRNPC facilitates the processing of pri-miR-21 [201], while hnRNPA2B1 promotes pri-miR-106b [202]. This modification not only boosts processing efficiency but may also facilitate nuclear export. NAT10-mediated ac^4^C on pri-miRNAs improves DGCR8 binding and promotes efficient processing. Elevated ac^4^C levels are found in some cancers, where this modification enhances miRNA maturation and contributes to altered gene expression [192].

Beyond internal modifications, both the 3′ and 5′ termini of miRNAs can undergo chemical alterations. At the 3′ end, tailing by TENTs introduces non-templated nucleotides—most commonly uridines or adenosines—onto pre- or mature miRNAs, which can either inhibit or promote their processing. For example, in pre-let-7 and miR-105 with a 1-nt 3′ overhang, mono-uridylation restores the canonical 2-nt overhang, thereby facilitating efficient DICER processing and miRNA biogenesis [203]. At the 5′ end, the monophosphate group is critical for AGO loading. Enzymes such as CLP1 restore 5′ phosphorylation of miR-34 in response to DNA damage, enabling its rapid activation [204]. Conversely, 5′-O-methylation by BCDIN3D neutralizes this charge and impairs DICER processing, acting as a gatekeeper in miRNA maturation [194].

### 7.2. Modifications Impacting miRNA Function and Stability

Beyond processing, RNA modifications can also directly influence miRNA function by altering seed sequences or modifying terminal ends. Most A-to-I editing sites fall within the seed sequences [205] and weaken complementarity with original targets, often resulting in reduced repression. At the same time, they can redirect the miRNA to novel targets, generating distinct regulatory outputs that can profoundly influence disease progression [206]. Reduced A-to-I editing of miR-376a in high-grade gliomas leads to an accumulation of the unedited form (miR-376a*A). This unedited form promotes glioma cell migration and invasion, whereas the edited form (miR-376a*G) suppresses these features. Specifically, unedited miR-376a*A targets RAP2A, promoting glioma cell invasion, while the edited miR-376a*G targets AMFR, suppressing invasion [207].

2′-O-methylation and m^6^A can enhance miRNA association with AGO or improve inhibition efficacy through enhanced RISC stability. For instance, 2′-O-methylated miR-21-5p by HENMT1 exhibits greater AGO2 affinity and resistance to exonucleases, promoting sustained silencing [191]. MiRNA stability is also strongly influenced by 3′ terminal modifications. 2′-O-methylation generally protects miRNAs from exonucleolytic degradation, whereas oligo-uridylation typically serves as a degradation signal. For instance, LIN28 binds the terminal loop of pre-let-7 and recruits TUT4/7 to induce oligo-uridylation (rather than mono-uridylation), leading to pre-let-7 degradation by the 3′–5′ exonuclease DIS3L2 [208,209]. LIN28-mediated inhibition of let-7 plays a crucial role in regulating developmental timing, pluripotency, and glucose metabolism. Notably, LIN28 is frequently overexpressed in various cancers, resulting in elevated expression of oncogenes such as RAS, MYC, and HMGA2, which are normally targeted by let-7 [65].

Reactive oxygen species (ROS)-induced 8-oxoG formation in miRNAs can disrupt target pairing and alter their functional roles. For example, oxidized miR-184 misrecognizes BCL-xL and BCL-w, leading to their downregulation and influencing apoptosis [193]. Specifically, 8-oxoG can be predominantly introduced at position 7 of miR-1, and this single modification is sufficient to induce cardiac hypertrophy in mice. Notably, targeted inhibition of 8-oxoG–modified miR-1 in mouse cardiomyocytes attenuates hypertrophy [210]. In gliomas and hepatocellular carcinoma, oxidative modifications of miRNAs such as let-7 or miR-122 can shift their activity from tumor-suppressive to oncogenic —or vice versa—depending on the extent of oxidation [211]. ROS, generated in response to extracellular or intracellular stress, can contribute to disease—including cancer and neurodegenerative disorders—through miRNA modifications.

RNA modifications introduce a dynamic and adaptable layer of control over miRNA biogenesis and function. These modifications not only diversify the regulatory potential of miRNAs but also link environmental cues and cellular states to gene silencing programs. As our understanding of the epitranscriptome deepens, targeting these chemical switches holds promise for diagnostics and RNA-based therapeutics, particularly in cancer, neurodegeneration, and immune disorders.

## 8. Advances in MiRNA Research and Challenges in MiRNA Therapeutics

Substantial progress has been made in miRNA research: numerous miRNAs have been identified across species, both canonical and non-canonical biogenesis pathways have been elucidated, and diverse mechanisms and targets have been characterized. These achievements contributed to the awarding of the Nobel Prize for miRNA research. However, compared with siRNAs—which were discovered earlier, received the Nobel Prize sooner, and have shown remarkable success in disease therapy—no miRNA-based therapy has yet advanced to phase III clinical trials or received FDA approval. This reflects both the complexity of miRNAs and the current gaps in our understanding of their biology. Recent discoveries have expanded our knowledge of miRNAs and other small RNAs, providing new insights for their potential clinical applications, including diagnostics and therapeutics.

### 8.1. Expansion of the Small RNA Landscape

For years, miRNAs were considered the predominant class of small regulatory RNAs. Although other types of small RNAs, such as tRNA-derived small RNAs (tsRNAs) [212] and rRNA-derived small RNAs (rsRNAs) [213], were occasionally detected in the early small RNA sequencing datasets, they were often dismissed as sequencing noise. However, biochemical analyses (e.g., PAGE gels) consistently reveal small RNA bands outside the canonical 20–24 nt miRNA range, particularly around 30–35 nt, where tsRNAs are enriched [214].

Traditional small RNA sequencing methods, which are biased toward RNAs with a 5′ phosphate and 3′ hydroxyl group, greatly favor miRNA detection. Emerging sequencing technologies, such as PANDORA-seq [215], overcome biases introduced by terminal and internal RNA modifications and have revealed a more complex small RNA landscape [214]. These studies demonstrate that housekeeping-derived RNAs, including tsRNA and rsRNA, could dominate the small RNA pools across most tissues and cell types [216]. These RNAs are also present in AGO immunoprecipitants [217], suggesting that they may function through RISC or participate in the regulation of RNAi.

This broader small RNA spectrum raises critical questions about the role of miRNAs. Given that the copy number of AGO proteins in mammalian cells is approximately 1.5 × 10^5^ molecules per cell [218], miRNA abundance typically ranges from a few thousand to tens of thousands per cell [219], depending on the cell type. In contrast, tsRNAs and rsRNAs can be present at levels ten times higher than miRNAs, suggesting that they may occupy a substantial proportion of AGO proteins. In many miRNA studies, synthetic miRNAs are often overexpressed in cells, which can exaggerate their function and even cause off-target effects, rendering the results potentially questionable.

The function of small RNAs requires comprehensive investigation, as exemplified by reproductive studies highlighting the role of miRNAs in embryonic development. Intriguingly, tsRNAs are extremely enriched in mature mouse sperm [220]. Embryonic injection of small RNAs from mature sperm can mediate transgenerational phenotypes, with different classes of small RNAs exerting distinct biological effects: tsRNAs induce metabolic disorders in offspring, miRNAs can trigger embryonic lethality, whereas injection of total small RNAs allows embryos to survive, producing offspring with phenotypes resembling those induced by tsRNAs alone [221]. Although the precise mechanisms remain to be elucidated [222], tsRNAs may act as nutrient sensors [223,224] and regulate miRNA activity by competing for AGO loading, thereby modulating miRNA-mediated target repression, whose dysregulation can lead to embryonic lethality. Additionally, tsRNAs are generally heavily modified, a feature required for transgenerational inheritance [225], which may also differentially impact RISC function. These findings suggest that housekeeping-derived small RNAs provide a previously underappreciated layer of regulation, indirectly influencing miRNA function.

TsRNAs and rsRNAs are evolutionarily conserved small RNAs [226] that have been found to be dysregulated in cancer [227], neurodegeneration [228], and immune disorders [229,230]. Studying them from the perspective of miRNA regulation could provide novel insights into the miRNA field and help explain the inefficiencies observed in miRNA-based therapeutics.

### 8.2. AGO-Independent MiRNA Functions

MiRNAs primarily exert their regulatory functions by associating with AGO proteins, leading to translational repression and mRNA degradation. However, emerging evidence indicates that miRNAs can also exert regulatory effects independent of AGO.

One example is miR-328, which functions as an RNA decoy by binding to hnRNP E2. Under normal conditions, hnRNP E2 binds to a C-rich region in the 5′ UTR of CEBPA mRNA, thereby inhibiting its translation. MiR-328 competes with CEBPA mRNA for hnRNP E2 binding and consequently relieves translational repression, promoting CEBPA expression during myeloid differentiation [231]. In another example, certain miRNAs, such as let-7, can interact with Toll-like receptors 7/8 (TLR7/8), key components of innate immunity [232]. These GU-rich miRNAs can act as ligands for TLR7/8, triggering pro-inflammatory responses [233]. Additionally, miRNAs can be secreted in extracellular vesicles (EVs) and subsequently modulate immune or neuronal activity in recipient cells [234]. When dysregulated, these mechanisms may contribute to autoimmune pathologies, particularly in the presence of elevated RBP-specific autoantibodies [235].

Additionally, some miRNAs appear to function as RNA aptamers, adopting specific conformations that enable them to bind directly to proteins. For example, miR-711 can bind ion channel TRPA1 extracellularly in sensory neurons and induce chronic itch [236], while miR-1 binds to cardiac potassium channel KIR2.1 in the heart and modulates electrophysiological properties [237]. These findings suggest that miRNA secondary or tertiary structures, beyond their sequence, confer diverse functional capacities.

Given the structural flexibility of RNA, miRNA functionality can extend beyond classical RNAi, which relies heavily on linear base pairing. Induced modifications can stabilize their structure, enabling highly specific targeting. Exploring miRNAs as aptamers opens the door to precise modulation of receptors or ion channels, broadening their potential applications beyond gene silencing.

### 8.3. Clinical Outlook and Challenges

Despite the vast body of research and promising preclinical data, the clinical application of miRNA therapeutics lags behind that of siRNAs [238,239]. Several obstacles hinder miRNA drug development. The first challenge is the mild and multi-targeted activity of miRNAs. Typically, miRNAs exert modest effects on numerous targets, which complicates therapeutic design and dose optimization. Moreover, their specificity requires careful validation. Identifying direct miRNA targets remains difficult, as interactions observed in vitro often fail to replicate in vivo, and many detected changes may instead represent indirect effects [240]. Like a coin with two sides, although miRNAs can simultaneously target multiple genes—complicating therapeutic control—this also provides the opportunity to regulate several pathogenic genes at once.

There are two major approaches to modulating miRNAs and their targets. One strategy involves introducing miRNA mimics, which can simultaneously reduce the expression of multiple genes, thereby producing therapeutic effects in certain diseases or at specific disease stages. Conversely, miRNA inhibitors (antagomiRs) can be used to suppress endogenous miRNAs, thereby restoring the expression of repressed genes and promoting disease recovery [238]. The first miRNA mimic drug to enter clinical trials was MRX34, a liposome-encapsulated miR-34a mimic, which began Phase I testing in April 2013 for advanced hepatocellular carcinoma and other cancers. While MRX34 represented a landmark step as the first miRNA mimic in human clinical trials, it was later halted due to severe immune-related adverse effects [241]. Another example is Miravirsen, which targets human miR-122, an essential factor for hepatitis C virus (HCV) replication. By sequestering miR-122, Miravirsen reduces viral load and was the first microRNA-targeted drug to advance into human clinical trials, highlighting the therapeutic potential of miRNA inhibition in infectious diseases [242].

In recent years, with the clinical application of mRNA and siRNA therapeutics, research on RNA delivery methods in vivo and on modulating their immunogenicity through chemical modifications has advanced rapidly [240,243]. These developments can also be applied to the field of miRNA-based therapeutics; however, miRNAs differ from other RNA drugs in their mechanisms of action, and further studies are required to evaluate whether such modification strategies would affect their functions [244].

Compared with the challenges faced by miRNAs in clinical therapeutics, they have shown greater potential in clinical diagnostics. MiRNAs have been found to be present in the biofluids, especially when encapsulated within EVs [245]. As mentioned earlier, miRNAs display strong cell- and tissue-specific expression patterns, and under both physiological and pathological conditions, they can exist in different isoforms. These variations are often highly sensitive to disease progression, making miRNAs effective diagnostic biomarkers [245]. For example, serum-derived miR-205-5p has been identified as a promising biomarker candidate, capable of distinguishing between patients with pancreatitis and those with pancreatic cancer, with a reported accuracy of 91.5% [246]. Nevertheless, compared with circulating DNA detection, current methods for circulating RNA still require improvement in both accuracy and practicality, including reliable approaches to sample storage [247].

In summary, miRNAs remain at the preclinical stage in both therapeutic and diagnostic applications. Although their intrinsic characteristics currently limit their therapeutic use within the existing disease classification frameworks, their high sensitivity to disease states and the presence of distinct variants offer greater clinical potential in diagnostics.

## 9. Conclusions and Outlook

Over three decades since their discovery, miRNAs have evolved from obscure genetic elements to central pillars of gene regulation, with far-reaching implications for biology and medicine. This review has traversed their intricate life cycle—from transcription and multi-step biogenesis to canonical and non-canonical functions, decay pathways, and epitranscriptomic modifications—underscoring their roles in development, disease, and emerging therapeutic paradigms. Key insights include the tissue-specific transcriptional control, regulatory hubs like the Microprocessor and DICER complexes, diverse RISC-mediated silencing mechanisms extending to nuclear and mitochondrial compartments, and dynamic modifications that fine-tune miRNA activity in response to cellular cues.

Yet, challenges persist: the expanding small RNA landscape reveals miRNAs as part of a broader regulatory network, where tsRNAs and rsRNAs may modulate their potency. AGO-independent functions, such as decoy roles and aptamer-like interactions, further diversify miRNA biology, while therapeutic hurdles—like mild multi-target effects, delivery inefficiencies, and immunogenicity—have stalled clinical progress compared to siRNAs. Nonetheless, miRNAs’ high sensitivity to disease states, tissue-specific expression, and isoform variations position them as superior diagnostic biomarkers, with potential in liquid biopsies for early detection.

Future advances hinge on refined sequencing technologies, structural biology, and delivery innovations borrowed from mRNA therapeutics. By addressing these gaps, miRNA research can unlock targeted interventions for cancer, neurodegeneration, and beyond, transforming preclinical promise into clinical reality.

## Figures and Tables

**Figure 1 biomolecules-15-01393-f001:**
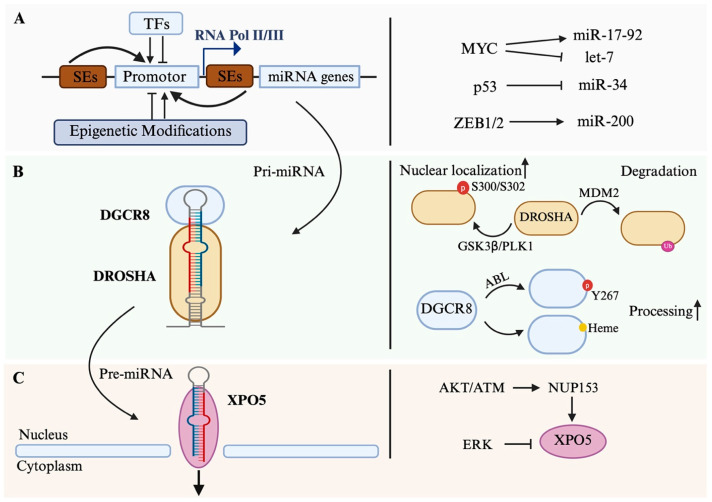
Regulation of miRNA biogenesis in the nucleus. (**A**) Transcriptional regulation of miRNA genes. Transcription factors (TFs) and super-enhancers (SEs), together with epigenetic modifications, modulate the activity of RNA polymerase II/III at miRNA promoters. Representative examples include MYC (induces miR-17–92 cluster and inhibits let-7 family, p53 (induces miR-34), and ZEB1/2 (inhibits miR-200). (**B**) Regulation of the Microprocessor complex. Pri-miRNAs are processed by DROSHA and its cofactor DGCR8. Representative examples of DROSHA/DGCT8 modifications: DROSHA nuclear localization is enhanced by phosphorylation at S300/S302 via GSK3β/PLK1, whereas MDM2-mediated ubiquitination promotes its degradation. DGCR8 activity is regulated by ABL-mediated phosphorylation at Y267 and heme binding, both of which promote pri-miRNA processing. (**C**) Nuclear export of pre-miRNAs. Exportin-5 (XPO5) transports pre-miRNAs from the nucleus to the cytoplasm. Representative examples of XPO5 regulation: XPO5 function is regulated by signaling pathways, including AKT/ATM via NUP153 and ERK-mediated phosphorylation.

**Figure 2 biomolecules-15-01393-f002:**
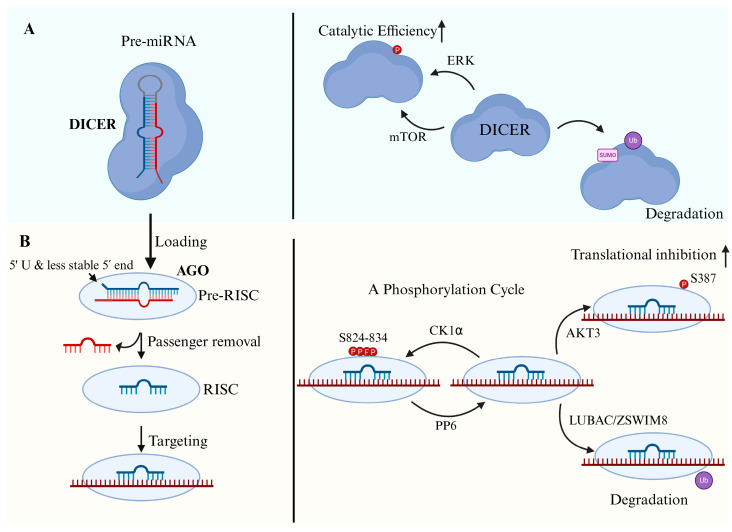
Regulation of miRNA biogenesis in the cytoplasm. (**A**) Regulation of DICER. Pre-miRNAs are processed by DICER to generate mature miRNA duplexes. DICER catalytic efficiency is enhanced by ERK- and mTOR-mediated phosphorylation. Conversely, post-translational modifications such as SUMOylation and ubiquitination promote DICER degradation, reducing miRNA processing capacity. (**B**) Regulation of AGO and RISC assembly. After DICER cleavage, duplex miRNAs are loaded onto AGO to form a pre-RISC. AGO preferentially selects the strand with a 5′ uridine and the less stable 5′ end as the guide strand. Subsequent removal of the passenger strand generates a mature RISC competent for target recognition. AGO function is dynamically regulated by post-translational modifications: (i) a CK1α–PP6 phosphorylation cycle at residues S824–834 controls target interaction and RISC remodeling; (ii) AKT3-mediated phosphorylation at S387 enhances miRNA-mediated translational inhibition; and (iii) ubiquitination mediated by LUBAC or ZSWIM8 promotes AGO degradation, reducing miRNA silencing activity.

**Figure 3 biomolecules-15-01393-f003:**
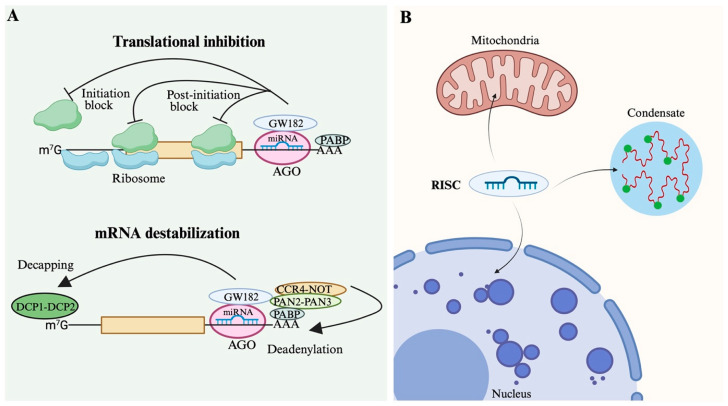
Mechanisms of miRNA-mediated gene regulation. (**A**) Translational inhibition and mRNA destabilization: Translational inhibition occurs when miRNA-AGO complexes block ribosome initiation or post-initiation, while mRNA destabilization involves decapping (via DCP1/DCP2) and deadenylation (via CCR4-NOT, PAN2-PAN3, PABP), leading to decay. (**B**) Subcellular localization of RISC: RISCs can function in the nucleus and mitochondria, and can also form condensates in the cytoplasm.

**Figure 4 biomolecules-15-01393-f004:**
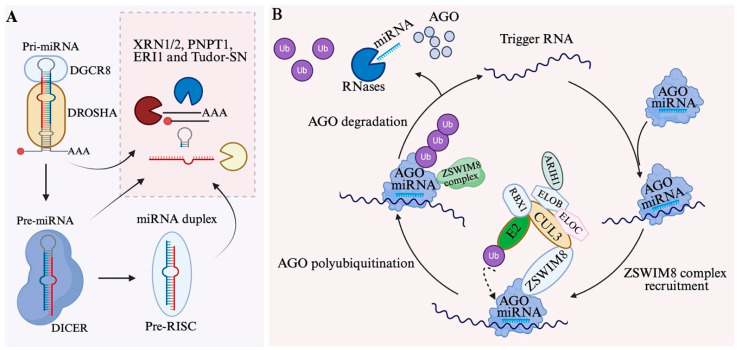
MiRNA decay mechanisms. (**A**) Byproducts from miRNA processing are degraded by exonucleases (XRN1/2, PNPT1, ERI1) and endonucleases (Tudor-SN). (**B**) ZSWIM8 mediated Target-directed miRNA degradation (TDMD): AGO-miRNA extended pairing with trigger RNA (e.g., mRNA or lncRNA) extracts the miRNA 3′ end from the AGO PAZ domain, stabilizing a distinct AGO conformation. This process, mediated by the ZSWIM8-Cullin-RING E3 ubiquitin ligase (ZSWIM8-CRL) complex (including ELOB, ELOC, CUL3, RBX1, and ARIH1), leads to AGO polyubiquitination and proteasomal degradation. The released miRNA is degraded by RNases, while the trigger RNA can bind a new AGO-miRNA complex, enabling multiple turnover-like activity and suprastoichiometric miRNA level reduction. Tailing or trimming of the miRNA 3′ end may occur but is not required for TDMD.

## Data Availability

Not applicable.

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
