# Peer review of "The Life of MicroRNAs: Biogenesis, Function and Decay in Cancer"

_biomolecules, 2025, doi:10.3390/biom15101393_

Round 1

Reviewer 1 Report

Comments and Suggestions for Authors

This is a very dense review that cover many aspects of miRNA biology. It is well organized covering several classical and new aspects of miRNA function. Nevertheless, sometimes the writing seem too colloquial and non-scientific standard. I would recommend a English proofreading.

Please find below some coments:

  • Abstract
    • Line11: microRNA significance is not in RNA interference. This is a distinct mechanism of posttranscriptional regulation. Please correct it all over the manuscript.
    • Line26: “dismissed as anomalies” is too colloquial. Please, rephrase this sentence.
    • Line 37: “a2-7nt seed sequence” is wrong because it means that seed sequence could have 2 -7nt long. In fact, seed sequence usually ranges from nt 2 to 8, and can be 5-6nt in length. Please rephrase to contain the correct concept of seed sequence.
    • Line 85: please, rephrase the sentence because it is confusing.
    • Line215: “miRNA precursors through compete pri-miRNA with” is confusing. Did you mean through competition? Please, correct the sentence.
    • Line 460: is it possible to describe the mechanism how AGO is directed to the nucleus coupled with mature miRNA?
    • Line 461: the sentence is confusing. “Histone methyltransferase euchromatic histone lysine methyltransferase 2 “
    • Line 498: this is not a RNAi mechanism. Please, correct to inform the concept related to microRNAs.
    • Line 510: “and their expression with associated with various cancer”. This sentence is confusing. Please, correct.
    • Line 511: this sentence is confusing: While AGO expression can be either upregulated or downregulated, consistent with other miRNA processing factors, this suggests that miRNA function is highly tissue- and stage-specific.
    • Line 541: processes such as tumor progression, neurological disorders. There should be an “and” instead of comma.
    • Line576: which call Target-Directed miRNA Degradation. Sentence needs correction
    • Line 609 and line 615: combine both paragraphs to indicate the role of ZSWIM8 before describing the effects of Ko model.
    • Line 628-629: As detection technologies continue to evolve [185], it is likely that many additional miRNA modifications remain to be uncovered. I think the sentence should conclude different: many additional miRNA modifications will be uncovered or discovered.
    • Line 635: And at least 6% of miRNAs. This refers to mature miRNAs? If so, it should be indicated.
    • Line 640: RNA modifications, both internal and terminal. Please rephrase to a more complete understanding. What is internal and terminal?
    • Line 717: and our incomplete understanding of them. The sentence is colloquial.
    • Line 723-724: tRNA and rRNA. Please provide the definition of each one upon first appearance in the text.
    • Line 766: this sentence is confusing or colloquial: MiRNA function has traditionally been understood through its association
    • Line 771: colloquial “lifting its translational repression”. Please explain in more detail how miR-328 derepress CEBPA
    • Line 804: the authors could mention one study where miRNAs were in clinical trial such as the miravirsen trial with miR-122 inhibition in HCV, and other trials with miRNA mimic/inhibitor
    • Line 822: I do not completely agree that the mechanism of action of miRNAs limit their therapeutical use. I think the authors should also bring a the other point of view that the pleiotropic effect could be interesting in the therapeutical aspect, specially for cancer. The main problem that is not cited is the delivery mechanism for any miRNA based therapy. I think it should be raised in a paragraph.
    • I think at some point the authors should also mention the concept of ceRNA (competing endogenous RNA) for some other RNAs that compete with miRNA such as lncRNA that may act as sponges, causing derepression of miRNA targets. Maybe in a paragraph while talking about tsRNA?
    •  
    •  
Comments on the Quality of English Language

Sometimes the writing seem too colloquial and non-scientific standard. I would recommend a English proofreading to detect this mistakes and correct them.

Author Response

To Dear Review,

We sincerely thank the reviewer for their thorough and constructive feedback. The insightful comments and suggestions have greatly helped us improve the clarity, scientific rigor, and overall quality of the manuscript. We appreciate the time and effort invested in carefully evaluating our work and providing detailed guidance, which has strengthened both the presentation and the content of our review.

Comment: The English writing is sometimes too colloquial and non-scientific.

Response: We thank the reviewer for this observation. We have thoroughly revised the manuscript for scientific tone and clarity and had the text professionally proofread.

Comment:Line11: microRNA significance is not in RNA interference. This is a distinct mechanism of posttranscriptional regulation. Please correct it all over the manuscript.

Comment:Line 498: this is not a RNAi mechanism. Please, correct to inform the concept related to microRNAs.

Response: We thank the reviewer for this important comment. RNA interference (RNAi) was discovered by Andrew Fire and Craig Mello in C. elegans, who shared the 2006 Nobel Prize in Physiology or Medicine. In their 1998 Nature paper (Fire et al., 1998), they showed that injection of double-stranded RNA targeting GFP mRNA inhibited GFP protein synthesis. Similar phenomena had also been observed before in plants (“post-transcriptional gene silencing,” PTGS) and in fungi (“quelling”). Mechanistic studies later established that Argonaute is the functional core of RNAi and that siRNAs can direct target RNA cleavage (Hammond, S M et al. Nature. 2000; Zamore, P D et al. Cell, 2000; Hammond, S M et al. Science, 2001) or translational inhibition (Zeng, Yan et al. PNAS, 2003; Doench, John G et al. Genes & development, 2003). And in the 2006 Nobel Prize Popular information released by The Nobel Committee for Physiology or Medicine, also mentioned miRNA that “What are the natural roles of RNA interference in organisms? We now know that several hundred genes in our genome encode small RNA molecules collectively called microRNA, which contain segments of codes from other genes… It is these molecules that activate the RNA interference machinery. The result is that mRNA from a gene with a similar coding sequence is broken down or blocked so that the protein cannot be produced.” Hence, as noted by Kim et al. (Kim, Haedong et al. Nature reviews. Molecular cell biology, 2025), “siRNAs are essentially artificial miRNAs.” Victor Ambros, upon receiving the 2024 Nobel Prize, also remarked: “…RNAi, I considered that to be an appropriate prize which encompassed microRNAs”. In our view, RNAi is a biological process in which RNA molecules mediate sequence-specific suppression of gene expression. Although siRNAs and miRNAs preferentially act through different mechanisms, both discoveries highlight the importance of RNAi as a powerful tool for regulating gene expression with strong therapeutic potential.

Comment: Line26: “dismissed as anomalies” is too colloquial. Please, rephrase this sentence.

Response: Rephrased to: "Initially overlooked as species-specific oddities".

Comment: Line 37: “a2-7nt seed sequence” is wrong because it means that seed sequence could have 2 -7nt long. In fact, seed sequence usually ranges from nt 2 to 8, and can be 5-6nt in length. Please rephrase to contain the correct concept of seed sequence.

Response: Corrected to: " followed by a seed sequence spanning nucleotides 2–8 (with possible variations down to 5–6 nt) that is complementary to target RNAs ".

Comment: Line 85: please, rephrase the sentence because it is confusing.

Response: Sentence has been rephrased for clarity.

Comment:Line215: “miRNA precursors through compete pri-miRNA with” is confusing. Did you mean through competition? Please, correct the sentence.

Response:Corrected. “NF90 can inhibit the processing of certain miRNA by binding to pri-miRNAs and ste-rically hinders the Microprocessor’s access to their cleavage sites”

Comment:Line 460: is it possible to describe the mechanism how AGO is directed to the nucleus coupled with mature miRNA?

Response: Thank you for the suggestion. We have expanded our discussion to include studies on AGO transport to the nucleus, thereby improving the completeness of our description of AGO regulation.

Comment:Line 461: the sentence is confusing. “Histone methyltransferase euchromatic histone lysine methyltransferase 2 “

Response: Corrected.

Comment:Line 510: “and their expression with associated with various cancer”. This sentence is confusing. Please, correct.

Response: Corrected.

Comment:Line 511: this sentence is confusing: While AGO expression can be either upregulated or downregulated, consistent with other miRNA processing factors, this suggests that miRNA function is highly tissue- and stage-specific.

Response: Rephrased for clarity.

Comment:Line 541: processes such as tumor progression, neurological disorders. There should be an “and” instead of comma.

Response: Corrected.

Comment:Line576: which call Target-Directed miRNA Degradation. Sentence needs correction

Response: Rephrased the sentence.

Comment:Line 609 and line 615: combine both paragraphs to indicate the role of ZSWIM8 before describing the effects of Ko model.

Response: The role of ZSWIM8 has been described at the beginning of Section 6.2. The text was rephrased to emphasize its importance before discussing knockout models.

Comment:Line 628-629: As detection technologies continue to evolve [185], it is likely that many additional miRNA modifications remain to be uncovered. I think the sentence should conclude different: many additional miRNA modifications will be uncovered or discovered.

Response: Corrected.

Comment:Line 635: And at least 6% of miRNAs. This refers to mature miRNAs? If so, it should be indicated.

Response: Corrected to: “at least 6% of mature miRNAs.”

Comment:Line 640: RNA modifications, both internal and terminal. Please rephrase to a more complete understanding. What is internal and terminal?

Response: Corrected to provide clearer explanation of internal (within the miRNA sequence) and terminal (at the 3′ or 5′ ends) modifications.

Comment:Line 717: and our incomplete understanding of them. The sentence is colloquial.

Response: Corrected to a more formal phrasing.

Comment:Line 723-724: tRNA and rRNA. Please provide the definition of each one upon first appearance in the text.

Response: We thank the reviewer for the comment. As tRNA (transfer RNA) and rRNA (ribosomal RNA) are widely recognized terms in molecular biology, we did not add definitions, considering the expertise of the intended readership of this journal.

Comment:Line 766: this sentence is confusing or colloquial: MiRNA function has traditionally been understood through its association

Response: Rephrased for clarity and scientific tone.

Comment:colloquial “lifting its translational repression”. Please explain in more detail how miR-328 derepress CEBPA

Response: Thank you for pointing this out. We clarified the mechanism.

Comment:Line 804: the authors could mention one study where miRNAs were in clinical trial such as the miravirsen trial with miR-122 inhibition in HCV, and other trials with miRNA mimic/inhibitor

Response: Added two examples of miRNA-based therapeutics in clinical trials, including miravirsen (miR-122 inhibition in HCV) and MRX34 (a miR-34a mimic).

Comment:Line 822: I do not completely agree that the mechanism of action of miRNAs limit their therapeutical use. I think the authors should also bring a the other point of view that the pleiotropic effect could be interesting in the therapeutical aspect, specially for cancer. The main problem that is not cited is the delivery mechanism for any miRNA based therapy. I think it should be raised in a paragraph.

Response: We thank the reviewer for this insightful comment. Our intention was to emphasize that, within the current “One-Molecule, One-Target, One-Disease” paradigm of clinical medicine, the pleiotropic effects of miRNAs may complicate drug development and the interpretation of therapeutic efficacy using conventional clinical endpoints. For this reason, we considered pleiotropy primarily as a limitation rather than an advantage. Regarding delivery, we note that many advances have been driven by siRNA- and mRNA-based therapies, which are expected to facilitate the clinical translation of miRNA therapeutics as well. Therefore, we did not discuss delivery in detail in the current manuscript.

Comment:I think at some point the authors should also mention the concept of ceRNA (competing endogenous RNA) for some other RNAs that compete with miRNA such as lncRNA that may act as sponges, causing derepression of miRNA targets. Maybe in a paragraph while talking about tsRNA?

Response: We thank the reviewer for this suggestion. We agree that miRNA sponges, including competing endogenous RNAs such as lncRNAs and circRNAs, are important regulators of miRNA function. We have added a paragraph in the section on Post-Translational Regulation of RISC to address this point.

Reviewer 2 Report

Comments and Suggestions for Authors

plese read the pdf enclosed.

Author Response

Comments and requests by the Referee on the manuscript draft – The life of MicroRNAs: Biogenesis, Function and Decay in Cancer- submitted by Shuang Din et al to Biomolecules.

I appreciated very much this review on MicroRNAs; the authors have deepen various items of the biogenesis, function and decay in Cancer and have included the most recent corresponding references. Further the authors have included a short summary at the end of each significant paragraph giving the reader the possibility to verify the understanding of the text. Many scientists are working on MicroRNAs, but on specific items and contexts, and this review is of major help to get an updated and elaborated summary of the biogenesis of the MicroRNAs.

I ask the authors to modify FIG.2B, because it is not clear the passenger strand removal and the use of guide strand microRNA: the text is clear (4.3), but not the Fig.2B. I want to present my compliments to the authors and I hope they will write in the next future an elaborated review on the extracellular microRNAs, that is on the secreted microRNAs mainly in EV.

Response: We sincerely appreciate the reviewer’s thoughtful comments and positive feedback. We have modified Figure 2B to clarify the process of passenger strand removal and the selection of the guide strand, ensuring consistency with the text in Section 4.3. We are also grateful for the suggestion regarding extracellular miRNAs and will consider this important topic for a future review.

Reviewer 3 Report

Comments and Suggestions for Authors

The manuscript presents a comprehensive review of microRNA (miRNA) biogenesis, function, and decay, with a focus on their role in cancer. The authors summarize transcriptional regulation, canonical and noncanonical processing pathways, RISC assembly, RNA modifications, miRNA decay mechanisms, and therapeutic potential. The topic is highly relevant to the field of molecular oncology and fits well within the scope of Biomolecules. However, the review requires substantial revisions to improve scientific depth, update the literature, and enhance its structure and clarity before it can be considered for publication.

The manuscript provides an extensive overview of miRNA biology, but it falls short in several key areas.

  • First, it lacks integration of recent discoveries from 2022–2025, including high-resolution cryo-EM structural insights into Drosha/DGCR8 and Dicer, advances in epitranscriptomic regulation (m⁶A, ac⁴C, A-to-I editing), and the increasing relevance of noncanonical biogenesis pathways such as mirtrons, snoRNA-derived miRNAs, and tsRNAs.
  • Second, while the title emphasizes cancer, the review does not sufficiently connect alterations in miRNA biogenesis with specific oncological contexts. Numerous well-characterized examples — such as DICER1 hotspot mutations in Wilms tumor, DGCR8/DROSHA defects in pediatric tumors, Exportin-5 (XPO5) alterations in hepatocellular carcinoma, AGO2 amplifications in colorectal cancer, and LIN28 overexpression suppressing let-7 in breast and gastric cancer — are missing or only superficially mentioned.
  • Third, the manuscript’s organization is suboptimal: canonical, noncanonical, cancer-related, and therapeutic aspects are intermingled, which makes the narrative difficult to follow. Finally, figures, tables, and reference lists require major improvements to bring the review to a publishable standard.

Detailed Comments

 a.- Scientific Completeness 

The manuscript omits several critical updates in the field:

  • Structural biology advances: Recent cryo-EM studies (2022–2023) on Drosha/DGCR8 and Dicer reveal structural determinants for substrate selection and cleavage fidelity, but these findings are not discussed.
  • Epitranscriptomic regulation: The review briefly mentions RNA modifications but does not integrate recent evidence on m⁶A-dependent pri-miRNA processing (METTL3/HNRNPA2B1), ac⁴C-mediated DGCR8 recruitment, and A-to-I editing effects on miRNA targeting, all of which have profound implications in cancer.
  • Noncanonical miRNA pathways: Emerging evidence on mirtrons, AGO2-dependent miR-451 maturation, snoRNA-derived miRNAs, and tsRNAs highlights their increasing relevance in tumor biology but is only superficially addressed.
  • IsomiRs: Recent studies (e.g., Wagner et al., 2024) demonstrate the functional and biomarker relevance of miRNA isoforms in cancer progression and therapy resistance; however, this is not discussed.

Incorporating these developments is essential to ensure that the review reflects the current state of knowledge.

b.- Cancer 

Despite the manuscript’s title, the cancer dimension is underdeveloped. The authors summarize miRNA biology in general but rarely link pathway alterations to specific cancer phenotypes. Examples that should be integrated include:

  • DICER1 mutations (RNase IIIb domain) → Wilms tumor, pleuropulmonary blastoma, ovarian cancers.
  • DROSHA/DGCR8 loss-of-function mutations → miRNA depletion in pediatric tumors.
  • XPO5 mutations → impaired nuclear export in colon cancer and HCC.
  • LIN28 overexpression → suppression of let-7 family, activation of RAS/MYC pathways.
  • AGO2 amplifications → dysregulated RISC activity in breast and colorectal cancers.

A dedicated table summarizing key biogenesis factors, their alterations, associated cancer types, and functional consequences would make the review more clinically relevant and reader-friendly.

c.- Organization and Structure

The manuscript’s organization could be improved to enhance readability. I recommend restructuring the review into the following logical sequence:

  1. Canonical miRNA biogenesis.
  2. Noncanonical pathways and their physiological relevance.
  3. Epitranscriptomic and RBP-mediated regulatory layers.
  4. Cancer-specific alterations in biogenesis pathways.
  5. Emerging therapeutic applications and biomarkers.

This structure would provide a clearer separation between mechanistic insights, cancer relevance, and translational potential.

d.- Figures and Tables

The current figures are schematic and insufficiently detailed. I recommend:

  • A comprehensive figure comparing canonical vs noncanonical miRNA pathways, including regulators, cofactors, and cancer-relevant checkpoints.
  • A dedicated figure explaining ZSWIM8-mediated target-directed miRNA decay (TDMD) using updated mechanistic insights.
  • A table summarizing RNA modifications (m⁶A, A-to-I, ac⁴C, 2'-O-methylation), their enzymes, effects on miRNA processing/stability, and associated cancer implications.

e.- Therapeutic Applications

The section on miRNA-based therapeutics is underdeveloped and should be expanded to cover:

  • Clinical progress on miRNA mimics (e.g., MRX34 for miR-34), antagomiRs, and nanoparticle-based delivery systems.
  • Lessons learned from failed clinical trials, including immune activation, off-target effects, and stability challenges.
  • Liquid biopsy applications using circulating miRNAs and isomiRs as cancer biomarkers.

f.- References and Language

The reference list is outdated, with limited coverage of studies after 2021. Additionally, careful proofreading is necessary to correct grammatical inconsistencies, improve sentence structure, and reduce redundancy.

Comments on the Quality of English Language

Careful proofreading is necessary to correct grammatical inconsistencies, improve sentence structure, and reduce redundancy.

Author Response

Comments and Suggestions for Authors

The manuscript presents a comprehensive review of microRNA (miRNA) biogenesis, function, and decay, with a focus on their role in cancer. The authors summarize transcriptional regulation, canonical and noncanonical processing pathways, RISC assembly, RNA modifications, miRNA decay mechanisms, and therapeutic potential. The topic is highly relevant to the field of molecular oncology and fits well within the scope of Biomolecules. However, the review requires substantial revisions to improve scientific depth, update the literature, and enhance its structure and clarity before it can be considered for publication. The manuscript provides an extensive overview of miRNA biology, but it falls short in several key areas.

First, it lacks integration of recent discoveries from 2022–2025, including high-resolution cryo-EM structural insights into Drosha/DGCR8 and Dicer, advances in epitranscriptomic regulation (m⁶A, ac⁴C, A-to-I editing), and the increasing relevance of noncanonical biogenesis pathways such as mirtrons, snoRNA-derived miRNAs, and tsRNAs.

Response: We thank the reviewer for this valuable suggestion. We fully agree that recent advances, including high-resolution cryo-EM structures of Drosha/DGCR8 and Dicer, epitranscriptomic modifications, and noncanonical biogenesis pathways such as mirtrons, are highly relevant to the field. However, given the scope and length limitations of this review, we chose to focus primarily on the canonical pathway of miRNA biogenesis. Each of these emerging areas is substantial enough to warrant dedicated reviews, and we believe that concentrating on the canonical pathway allowed us to provide a more focused and coherent narrative. Nevertheless, we acknowledge their importance and have added brief mentions and citations where appropriate to guide readers toward these developments.

Second, while the title emphasizes cancer, the review does not sufficiently connect alterations in miRNA biogenesis with specific oncological contexts. Numerous well-characterized examples —such as DICER1 hotspot mutations in Wilms tumor, DGCR8/DROSHA defects in pediatric tumors, Exportin-5 (XPO5) alterations in hepatocellular carcinoma, AGO2 amplifications in colorectal cancer, and LIN28 overexpression suppressing let-7 in breast and gastric cancer—are missing or only superficially mentioned.

Response: We thank the reviewer for this important comment. Our intention was to provide a broad overview of the microRNA life cycle, emphasizing biogenesis, function, and decay, while highlighting their relevance to cancer biology. Given the scope and length limitations of this review, we did not aim to provide an exhaustive catalog of all cancer-specific alterations in miRNA processing factors. Instead, we focused on representative examples that illustrate how dysregulation of miRNA pathways contributes to oncogenesis. We selected representative cases of mutations and dysregulation in regulatory factors at each biogenesis layer to provide illustrative insights. While a detailed discussion of every mechanism exceeds the scope of the present review, we acknowledge their significance and have added citations and brief mentions to guide readers to the relevant literature.

Third, the manuscript’s organization is suboptimal: canonical, noncanonical, cancer-related, and therapeutic aspects are intermingled, which makes the narrative difficult to follow.

Response: Our manuscript’s structure is based on several Nature Review articles, upon which we have added recent research findings as well as our own perspectives.

Finally, figures, tables, and reference lists require major improvements to bring the review to a publishable standard.

Response: We consulted numerous articles from this journal to ensure that the figures are clear and concise, effectively supporting the main text of the manuscript.

Detailed Comments

 a.- Scientific Completeness

The manuscript omits several critical updates in the field:

  • Structural biology advances: Recent cryo-EM studies (2022–2023) on Drosha/DGCR8 and Dicer reveal structural determinants for substrate selection and cleavage fidelity, but these findings are not discussed.

Response: We have cited key recent cryo-EM studies, including those published in 2024. We have discussed substrate selection and cleavage fidelity by DROSHA/DGCR8 and DICER, focusing primarily on regulatory mechanisms rather than detailed structural changes, which could lead to an extensive and potentially endless discussion.

  • Epitranscriptomic regulation: The review briefly mentions RNA modifications but does not integrate recent evidence on m⁶A-dependent pri-miRNA processing (METTL3/HNRNPA2B1), ac⁴C-mediated DGCR8 recruitment, and A-to-I editing effects on miRNA targeting, all of which have profound implications in cancer.

Response: We have cited several of these studies in the manuscript..

  • Noncanonical miRNA pathways: Emerging evidence on mirtrons, AGO2-dependent miR-451 maturation, snoRNA-derived miRNAs, and tsRNAs highlights their increasing relevance in tumor biology but is only superficially addressed.

Response: We focused primarily on the canonical miRNA pathway in this review. However, we have added an independent Section 8.1 to discuss other small RNAs, such as tsRNAs, which we believe are increasingly important in the context of miRNA research.

  • IsomiRs: Recent studies (e.g., Wagner et al., 2024) demonstrate the functional and biomarker relevance of miRNA isoforms in cancer progression and therapy resistance; however, this is not discussed.

Response:There more than 10000 miRNA paper each year, we focus more on the biogenesis canonical pathway. Wagner et al., 2024 is a review about isomiR, we cited in the revised manuiscript.  

Incorporating these developments is essential to ensure that the review reflects the current state of knowledge.

b.- Cancer

Despite the manuscript’s title, the cancer dimension is underdeveloped. The authors summarize miRNA biology in general but rarely link pathway alterations to specific cancer phenotypes. Examples that should be integrated include:

  • DICER1 mutations (RNase IIIb domain) → Wilms tumor, pleuropulmonary blastoma, ovarian cancers.
  • DROSHA/DGCR8 loss-of-function mutations → miRNA depletion in pediatric tumors.
  • XPO5 mutations → impaired nuclear export in colon cancer and HCC.
  • LIN28 overexpression → suppression of let-7 family, activation of RAS/MYC pathways.
  • AGO2 amplifications → dysregulated RISC activity in breast and colorectal cancers.

A dedicated table summarizing key biogenesis factors, their alterations, associated cancer types, and functional consequences would make the review more clinically relevant and reader-friendly.

Response: We thank the reviewer for this valuable suggestion. The cancer-related miRNA field contains many controversial and evolving studies; therefore, we focused on citing and discussing well-validated work. We have highlighted key biogenesis factors and provided examples of how their dysregulation contributes to cancer. While a dedicated table summarizing all factors and associated cancers could be useful, several excellent reviews have already provided such summaries, which we have cited in the manuscript. Hence, we chose to emphasize mechanistic insights rather than replicate existing tables.

c.- Organization and Structure

The manuscript’s organization could be improved to enhance readability. I recommend restructuring the review into the following logical sequence:

  1. Canonical miRNA biogenesis.
  2. Noncanonical pathways and their physiological relevance.
  3. Epitranscriptomic and RBP-mediated regulatory layers.
  4. Cancer-specific alterations in biogenesis pathways.
  5. Emerging therapeutic applications and biomarkers.

This structure would provide a clearer separation between mechanistic insights, cancer relevance, and translational potential.

Response: Our manuscript’s structure is based on several Nature Review articles, upon which we have added recent research findings as well as our own perspectives.

d.- Figures and Tables

The current figures are schematic and insufficiently detailed. I recommend:

  • A comprehensive figure comparing canonical vs noncanonical miRNA pathways, including regulators, cofactors, and cancer-relevant checkpoints.
  • A dedicated figure explaining ZSWIM8-mediated target-directed miRNA decay (TDMD) using updated mechanistic insights.
  • A table summarizing RNA modifications (m⁶A, A-to-I, ac⁴C, 2'-O-methylation), their enzymes, effects on miRNA processing/stability, and associated cancer implications.

Response: We appreciate the reviewer’s suggestions. Our review primarily focuses on the canonical miRNA pathway. We have included an updated figure and citations for ZSWIM8-mediated target-directed miRNA decay (TDMD), which we believe sufficiently illustrate the mechanism of miRNA decay. For RNA modifications and noncanonical pathways, we have highlighted key aspects in the text and cited relevant studies, while maintaining a clear and concise figure presentation to avoid excessive complexity.

e.- Therapeutic Applications

The section on miRNA-based therapeutics is underdeveloped and should be expanded to cover:

  • Clinical progress on miRNA mimics (e.g., MRX34 for miR-34), antagomiRs, and nanoparticle-based delivery systems.
  • Lessons learned from failed clinical trials, including immune activation, off-target effects, and stability challenges.
  • Liquid biopsy applications using circulating miRNAs and isomiRs as cancer biomarkers.

Response: We thank the reviewer for the suggestion. In the revised manuscript, we have expanded the discussion on miRNA-based therapeutics by including detailed examples such as MRX34 and Miravirsen, highlighting their clinical progress, mechanisms, and challenges.

f.- References and Language

The reference list is outdated, with limited coverage of studies after 2021. Additionally, careful proofreading is necessary to correct grammatical inconsistencies, improve sentence structure, and reduce redundancy.

To reviewer: We sincerely thank the reviewer for the thorough and constructive feedback. We greatly appreciate the recognition of the relevance and scope of our review, as well as the detailed suggestions for improving scientific depth, organization, and clarity. In response, we have cited over 50 papers published after 2020, including several from 2025. We also refined the organization, updated figures, and added clarifying explanations to enhance readability and scientific rigor. The reviewer’s comments have been invaluable in strengthening the manuscript, and we are grateful for the time and expertise dedicated to providing such comprehensive guidance.

Reviewer 4 Report

Comments and Suggestions for Authors

Dear authors, The review is devoted to a detailed description of the life cycle of miRNAs from their formation in the nucleus to degradation in the cytoplasm. The review focuses on what disturbances in the functioning of miRNAs can be associated with different types of cancer. The text is written in simple and understandable language and contains many details necessary for understanding the role of miRNAs in the life of the body.
There is a typo on line 262 - the name of the figure is incorrectly indicated.
Further comments are of a recommendatory nature.
The text contains a significant amount of information, as well as many gene names and abbreviations. The list of abbreviations is given only at the end, which somewhat complicates the perception of the text. Moreover, it does not cover all the abbreviations used.
Perhaps it is necessary to add a figure illustrating section 4.2, which would also facilitate perception and allow a more complete understanding of the mechanisms of action of TRBP and PACT.
Also, there is not enough emphasis on oncological diseases, as indicated in the title of the review. In particular, in many cases, data are provided characterizing many other diseases. And it is difficult to argue with this, since disruption of miRNA functioning at any stage will inevitably lead to significant dysregulation of all processes in the cell, but will not necessarily lead to its malignant transformation.

Author Response

Comments and Suggestions for Authors

Dear authors, The review is devoted to a detailed description of the life cycle of miRNAs from their formation in the nucleus to degradation in the cytoplasm. The review focuses on what disturbances in the functioning of miRNAs can be associated with different types of cancer. The text is written in simple and understandable language and contains many details necessary for understanding the role of miRNAs in the life of the body.
There is a typo on line 262 - the name of the figure is incorrectly indicated.

Response: We sincerely appreaciate reviewer’s commends. We corrected the typo.  

Further comments are of a recommendatory nature.
The text contains a significant amount of information, as well as many gene names and abbreviations. The list of abbreviations is given only at the end, which somewhat complicates the perception of the text. Moreover, it does not cover all the abbreviations used.
Perhaps it is necessary to add a figure illustrating section 4.2, which would also facilitate perception and allow a more complete understanding of the mechanisms of action of TRBP and PACT.

Response: We thank the reviewer for the suggestion. We have added the previously missing abbreviations to the list at the end of the manuscript. Regarding the mechanisms of TRBP and PACT in DICER regulation, these processes are highly complex. To avoid overcomplicating the figure and distracting from the main message, we have chosen not to include them in the schematic.

Also, there is not enough emphasis on oncological diseases, as indicated in the title of the review. In particular, in many cases, data are provided characterizing many other diseases. And it is difficult to argue with this, since disruption of miRNA functioning at any stage will inevitably lead to significant dysregulation of all processes in the cell, but will not necessarily lead to its malignant transformation.

Response: We thank the reviewer for the comment. While miRNAs are implicated in a wide range of diseases, our review primarily focuses on their biogenesis, with an emphasis on oncological contexts where these mechanisms are most extensively studied. To provide a broader perspective, we have also included selected examples from other disease contexts, such as neurodevelopmental and autoimmune disorders, highlighting the relevance of miRNA regulation beyond cancer.

To reviewer:

We sincerely thank the reviewer for their thoughtful and constructive feedback. Their comments regarding figure clarity, abbreviation coverage, and emphasis on oncological contexts have been invaluable in improving the clarity, focus, and scientific rigor of our manuscript.

Round 2

Reviewer 3 Report

Comments and Suggestions for Authors

Dear Authors

I appreciate your detailed answers and the improvements of the manuscript. I am satisfied with your work, that has improved the manuscript contents, figures and outline.

I would recommend the article for publication

Sincerely